# Efficient Secure Communication in Zigbee Network Using the DNA Sequence Encryption Technique

**DOI:** 10.3390/life13051147

**Published:** 2023-05-09

**Authors:** Bhukya Padma, Erukala Suresh Babu

**Affiliations:** Department of CSE, NIT Warangal, Warangal 506004, Telangana, India

**Keywords:** Zigbee network, DNA sequence, Internet of Things, secure communication

## Abstract

Zigbee IoT devices have limited computational resources, including processing power and memory capacity. Therefore, because of their complicated computational requirements, traditional encryption techniques are inappropriate for Zigbee devices. Because of this, we proposed a novel, “lightweight encryption” method (algorithm) is based on “DNA sequences” for Zigbee devices. In the proposed way, we took advantage of the randomness of “DNA sequences” to produce a full secret key that attackers cannot crack. The DNA key encrypts the data using two operations, “substitution” and “transposition”, which are appropriate for Zigbee computation resources. Our suggested method uses the “signal-to-interference and noise ratio (SINR)”, “congestion level”, and “survival factor” for estimating the “cluster head selection factor” initially. The cluster head selection factor is used to group the network nodes using the “adaptive fuzzy c-means clustering technique”. Data packets are then encrypted using the DNA encryption method. Our proposed technique gave the best results by comparing the experimental results to other encryption algorithms and the metrics for energy consumption, such as “node remaining energy level”, key size, and encryption time.

## 1. Introduction

DNA sequencing is a growing area of research in many fields, such as IoT networks, forensic science, modern agriculture, and current medicine [1]. DNA sequencing requires an effective and quick algorithm because DNA sequences utilize large databases. “Adenine (A), guanine (G), cytosine (C), and thymine (T) make up the DNA (T)”. “Base + Sugar + Phosphate = Nucleotide”. Each pair of the chemical bases is joined by a sugar and a “phosphate molecule”. The advantages of computing the DNA structures, such as its enormous “parallelism” and “storage capacity” and “low-level power consumption”, have also been utilized for the development of novel cryptography techniques [1]. DNA sequencing refers to finding the exact nucleotide order (different sequences of the “letters A, G, C, and T”) within DNA molecules. Diseases show a certain pattern of DNA that can be detected by pattern matching. DNA is being extracted from organisms in exponentially more significant amounts. Therefore, pattern-matching techniques are crucial in many computational biology applications for data analysis involving proteins and genes. Their main goal is identifying a specific pattern within a given DNA sequence. 

### 1.1. Deoxyribonucleic Acid (DNA)

Deoxyribonucleic acid, also known as DNA, is a large molecule that houses the different genetic codes of each organism. It contains directions for generating every protein in our bodies, much like a recipe book. The four fundamental “bases” that make up “DNA are adenine (A), cytosine (C), guanine (G), and thymine (T)” [2]. These bases’ arrangement, or sequence, forms the genome’s instructions: a two-stranded molecule, DNA. DNA is shaped like a twisted ladder with a unique “double-helix” shape. Each double-helix strand of DNA is used as a template for reproducing the base sequence. Cells split, reproducing their DNA. This is crucial because each new cell requires an identical duplicate of the old cell’s DNA. DNA is responsible for conducting the information needed to build and run a particular organism. Certain parts of DNA are in charge of turning genes on and off, controlling the amount of the specific protein produced.

### 1.2. DNA Sequencing

DNA sequencing indicates the arrangement in which letters are combined in DNA. Individual genes, entire DNA, complete chromosomes, and whole genomes are sequenced through DNA sequencing. To sequence DNA, it must go through a lengthy scientific computing procedure. DNA computation is a complicated process involving a lot of precision and complex logic [3]. The state of DNA computation is still being worked out. DNA calculations have not yet surpassed the computing capabilities of current computers. We acquire sequenced DNA in a solution after the final phase of detection and reading, and then, the order of the base pairs that make up the strand of DNA is identified.

### 1.3. Disease Diagnosis Using DNA Sequencing

Disease diagnosis by DNA sequencing is a challenging medical subject to master. Diseases are caused by duplicating, repetitive, deleted, and disease-affected genes, among other factors. DNA sequencing is a first-stage diagnostic for disease detection that is perfectly safe.

DNA databases are massive and growing day after day. As a result, the analysis of datasets is not a manual process. The most common categories are exact, inexact, single, and multi-faceted model-matching algorithms. Model correspondence techniques’ main task is identifying known models from massive datasets [4]. Some inaccuracies may be regarded as per the application’s need in the inexact pattern-matching process, but “exact pattern-matching” has no errors recognized in the outcome. When there are several match patterns to scan for, a “single pattern matching algorithm” executes more than one string at a single instance. Users will find it more difficult to get critical information from sequences as the bulk of the data expands. As a result, faster pattern-matching algorithms require more efficient and resilient procedures. It is one of the most significant areas in bioinformatics that has been researched. The method is designed to get better as the dataset expands. Because bioinformatics requires precise findings, accurate and “multi-string pattern matching” techniques are utilized. A “single pattern matching algorithm” passes one string at an instance, and “multi-string patterns” to be explored pass more than one in a “single pass”.

The “Internet of Things (IoT)” results from numerous empowering technological advancements for “gathering”, “processing”, “inferring”, and “transmitting the data”, “including embedded systems”, “wireless sensor networks”, “cloud computing”, and “big data”. The IoT design comprises three layers: the “perception layer”, the “network layer”, and the “application layer”, which includes “RFID”, “WSN”, “sensors”, “readers”, “I.P. Cams”, “MEMS”, and other “devices” [5]. However, as the number of sensors within the “IoT framework” grows, the problem of transferring data between those devices becomes more complex, necessitating working considerations and foundation costs to offset data transfer requirements. Zigbee networks (Z.N.s) are essential to advance the Internet of Things, and several innovations have recently been implemented to make their integration easier. Z.N. distributions may be used to support the sensory capabilities required by future applications, so their integration with the Internet may significantly enhance its engineering. The ZnS is a well-established part of the “Internet of Things”. The IoT needs new technologies and established practices to connect the sensors and actuators that make up a Z.N. The Z.N. responds to problems in many spheres of life with innovative and potent solutions. Since Z.N.s play a significant role in enhancing daily life, developing low-effort, remote sensor systems hold tremendous research interest. The future Internet intends to coordinate various wired and wireless communication advancements to advance the IoT concept. Contrary to popular belief, Z.N.s are self-organizing networks of small, simple Zigbee devices communicating via multiple hops to provide screen and control functionality.

To guarantee “connectivity in sensor systems” and on Z.N.s, IoT systems rely on remote connections. “IoT gateways (IGWs)” enable the linking of various sensor types (“IoT gadgets”) and the IoT cloud using multiple technologies, such as “802.11a/b, Bluetooth”, “Bluetooth low energy”, and “Zigbee”. Organizing and explicitly directing the system is a fundamental IoT driving force that enables devices to connect. It entails creating traffic plans and sending the steered packages through a design from the source to the specified goal. Applications for detecting and observing in the natural, contemporary, and biomedical fields that rely on continuous data have significantly advanced the ongoing development of Z.N.s in the IoT. Some key qualities should be considered when structuring IoT devices for executives to adapt to new challenges, such as the limited resources of remote sensor devices distributed in organized conditions and enormous information gathered during various utilizations.

Our Contribution: This paper presents the proposed work by designing efficient and effective secure communication in the Zigbee network using DNA sequence encryption. There are a lot of inherent security flaws in a Zigbee network; therefore, attackers can join the Zigbee network and have chances to access data.

1.A novel “DNA block cipher symmetric algorithm” is used as the “DNA-based” encryption method for the secure routing of Zigbee devices in IoT networks [2,6,7]. A secret encryption key is generated using the DNA sequence (SecKey).2.The randomly generated SecKey is used throughout the encryption. Two robust and optimized operations, “substitution” and “transposition”, are applied to encrypt the data. This technique perfectly fits the “limited computation resources” of “Zigbee devices”. The encryption technique performs two steps: (1) key generation and (2) encryption.3.Each DNA sequence of any size generates the keys for the suggested algorithms. Each DNA sequence consists of alphabets, and a randomly arranged quartet of four letters forms a group. In Table 1, a letter represents 2 bits. The four DNA letters (A, C, G, and T) are then used to segment the DNA sequence into 1-byte segments (K1, K2,… Kn), where these segments stand in for the secret key (SecKey).4.The suggested DNA-enabled technique is a “block cipher” that substitutes and rearranges the input data’s bytes. The input data must be divided into N segments (B1, B2, …, Bn) with a segment size of 1 byte each before applying the substitution phase.

**Table 1 life-13-01147-t001:** Comparative Analysis of Existing Works with Our Contribution.

S.No.	Title	Zigbee	DNA Sequence	Internet of Things	Attacks	Security
1.	Manu Elappilaa et al. [8]	✓	×	✓	×	✓
2.	Al-Turjman et al. [9]	✓	×	✓	✓	✓
3.	A. Memos et al. [10]	×	×	✓	✓	✓
4.	Slavica Tomovic et al. [11]	✓	×	✓	×	✓
5.	Raju Bhukya et al. [12]	×	✓	×	×	✓
6.	Boyer–Moore et al. [13]	×	✓	×	×	✓
7.	Knuth–Morris–Pratt et al. [14]	×	✓	×	×	×
8.	S. Hasib, M. Motwani et al. [15]	×	✓	×	✓	✓
9.	Our Contribution	✓	✓	✓	✓	✓

× Represents feature will not support, ✓ Represents feature will support, Our proposed work will support above features.

## 2. Related Work

The “Agrep” algorithm for approximate nucleotide sequence matching was put forth by Hongjian Li et al. The current pattern-matching algorithms can only match a limited number of genome sequences [16,17]. A sequence matching algorithm that operates on discovering a wide range of genome patterns of length up to 64 bits with an adaptability of 9 bits was achieved by the “CUDA” implementation of “Agrep algorithms”. Its memory requirement is decreased using a binary representation with two bits per character. Consequently, multiple genomes can be loaded at an instance. Because the entire genome is strictly scanned, its high sensitivity is ensured [18]. The high sensitivity of the CUDA agreement—a measure of “the proportion of actual positives” identified correctly—is another benefit of the agreement. A novel “exact multiple pattern” matching algorithm utilizing “DNA sequence” and “pattern pair” was introduced by Raju Bhukya and DVLN Somayajulu [12]. Each pattern algorithm starts from the character that matches the pattern, reducing the need to compare additional characters if the indexes for the input sequence have already been created. It works well for applications involving DNA sequences; in some instances, the number of comparisons of the proposed algorithm decreases as the pattern size increases. The algorithm proposed in this paper outperforms other commonly used algorithms regarding comparisons and the “comparations per character (CPC)” ratio. The proposed algorithm outperforms many competing algorithms, including MSMPMA, brute force, TriMatch, naive string matching, and IKPMPM, regarding metrics such as CPC ratio. Comparing the EPMSPP approach to other, more traditional methods, it performs excellently.

In [13], the Boyer–Moore algorithm conducts a more significant shift increment when a mismatch is identified. In terms of scanning, it differs significantly from the naive algorithm. It traverses the string going right to the left, unlike the naive algorithm, in which the last character of P is matched with the first character of T. If a character matches, the pointer is advanced to the left to the pattern’s remaining elements. If a conflict is found in T at character c that does not exist in P, P is advanced right by m places and adjusted to the next character following c. P is moved exactly such that c is aligned with the rightmost appearance of c in P if the c is part of P. However, the worst-case time complexity remains O(m+n). M is first generated for the specified pattern P in the Knuth–Morris–Pratt finite state automata (FSM) model [14]. If a pattern exists in the text, it is approved or ignored. The KMP algorithm initially generates a supplementary LPS of size m (the same as the pattern size) to skip characters during matching. LPS represents the longest prefix suffix. It also serves as a suffix. KMP can know some of the following characters in the next window’s text. Knowing this, we can avoid matching characters we already know will match. The sole drawback of the KMP method is that it does not indicate the number of times the pattern has occurred. Dynamic programming is one of the highly utilized algorithms in computer science. It entails recursively addressing consecutive recurrence relations, in which more minor issues are addressed to answer the more significant problem. The worst time complexity is O(n).

According to S. Hasib et al. [15]. They proposed the “Aho–Corasick string matching algorithm”,, especially for real-world applications. The Aho–Corasick algorithm works well for multiple pattern matching and has many applications. However, it has been found that the algorithm’s performance suffers in terms of “time” and “space” as the size of the automata dramatically increases. The algorithm’s complexity grows linearly with the patterns’ scope, the text being searched, and the number of output matches. This algorithm can resolve issues in text mining, bioinformatics, digital forensics, intrusion detection, and plagiarism detection, among other things. The intrusion detection technique uses an intrusion detection system (IDS) to find intrusions. Burton Howard Bloom created the space-effective probabilistic data structure called the Bloom filter 1970. False eight-match positives can happen, but false negatives cannot. The result of a query is either “possibly in the set” or “definitely not in the set”, in other words. The number of items added affects the likelihood of false positives; elements can be added to the set but not removed (although the counting Bloom filter variant can address this issue). In general, the Bloom filters suffer from generating false positives. However, the space complexity will be reduced compared to other data structures, including hash tables and linked lists, because they do not store the actual data items. Pointers add additional linear space overhead to related systems.

A flexible “service-based” applications scheme for smart cities was developed by Fadi Al-Turjman et al. [9]. They were using a large volume of multimedia data. They also have a robust “mathematical model” for data packet routing. The recommended approach ensures “quality of service (QoS)” in “multimedia security” and “safety applications” while allowing for data routing on available vehicle assets. To determine the proposed model’s applicability, they conducted an “analytical study” to “validate the simulation” results regarding the “packet received ratio”, energy usage, and “average end-to-end delay”. An upcoming Internet of Things network design and the security issues it raises were presented by Vasileios A. Memos et al. [10]. in their article. Alireza Esfahani et al. proposed a “lightweight authentication solution” for “M2M communications in an Industrial IoT environment” based solely on “hash and XOR operations”. The proposed mechanism achieves “mutual authentication”, “session key agreement”, “device identity confidentiality”, and “resistance to ancillary attacks such as replay attacks”, “man-in-the-middle attacks”, “impersonation attacks”, and “modification attacks” while having a low “computational cost”, “communication overhead”, and “storage overhead”. Software-defined networking denotes “sophisticated traffic control” and “resource management components” in a coherently concentrated system control plane [8]. Fog computing, in contrast, enables small amounts of data to be analyzed and monitored at the “network edge”, supporting the applications that require “extremely low” and “predictable latency”. They also assessed the benefits and potential services of the proposed design, as did Manu Elappilaa.

As a low-energy routing strategy for WSNs, “survivable path routing” was developed. This protocol should work in high-traffic environments, such as those used by “IoT applications for remote healthcare monitoring”, when “many sources” simultaneously attempt to deliver packets to the exact location. The next hop of the node is decided by the four factors, including the “signal-to-interference ratio”, “noise ratio of the link”, “survival factor of the path”, and “congestion level at the next-hop node”.

## 3. Proposed Work

This section presents our proposed work, the Zigbee network head choosing factor from the start evaluated by the signal-to-interference ratio of Zigbee nodes, congestion level, and survivability factor of nodes [8]. Zigbee nodes are clustered utilizing adaptive fuzzy c-means clustering dependent on the cluster head choosing factor. After that, data packets are encrypted by using the proposed DNA-based encryption algorithm. Finally, an optimized route is obtained by adaptive krill herd optimization. Figure 1 depicts the proposed algorithm’s flow diagram.

Our “DNA-based encryption technique” for enabling the secure routing for Zigbee-enabled IoT devices uses a “DNA block cipher symmetric algorithm”. The “DNA sequence” generates a secret encryption key (SecKey). The two robust and optimized operations, “substitution” and “transposition”, are applied to encrypt the data. This proposed technique is perfectly suitable for devices with limited “computation resources”, such as Zigbee devices. Initially, “the cluster head choosing factor” is determined by the significant factors, including the “signal-to-interference ratio”, “congestion level”, and “survivability factor” of the Zigbee nodes [19,20]. Based on this evaluation, “network nodes” are clustered using adaptive “fuzzy c-means clustering”. Then, the DNA sequence encryption algorithm is used to send data securely.

The Zigbee network is made up of a set of nodes called $N_{O}$ = $\{N_{O_{1}}$, $N_{2}$, $N_{O_{3}}$ … $N_{O_{i}}$} and assumes the “source” as $N_{O_{1\;}}$ and the “destination” as $N_{O_{i}}$. The proposed study groups network nodes based on three efficient parameters. The suggested design process is thoroughly covered in the sections that follow.

A. Three Factors Used to Select Nodes: Three factors affect clustering nodes and are covered in this section: the links are the “signal-to-interference and noise ratio”, the “path’s survivability factor”, and the “congestion level at the next-hop node”.

### 3.1. Survivability Factor

The calculated ratio of “the minimal energy left between each node to the total energy needed for communication over the network” is known as the “survival factor [20,21]”. The “path survivability factor” is the “ratio of the total energy consumed” and the smallest amount of power that can be transferred between nodes. Condition (1) refers to this percentage:(1)FS=MfEf

Here, $E_{f}$ Is path L’s overall energy consumption and $M_{f}$ Is path L’s nodes’ minimum power value that is still available?

### 3.2. Signal Interference and Noise Ratio (SINR)

The “SINR” is calculated by the ratio of the “quality of the transmitted signal” and total “interference” and the background noise. The receiver’s $r_{e_{i}}$ level of noise and interference, designated by the transmission edge $e_{d_{i}}$, is given as follows:(2)$$I_{f}\;(e_{d_{i}})=\sum_{\begin{array}{c} m,m=1 \end{array}}G\left(T_{e_{m,}}{\;r}_{e_{i}}\right)p\left(T_{e_{m}}\;\right)+\lambda_{i}$$

Here, G $\left(T_{e_{m,}}{\;r}_{e_{i}}\right)$ is the “path gain between the transmitter”. $T_{e_{m,}}\;$ on the current link, m e, and the “receiver”. $r_{e_{i}}$ on the current edge $e_{d_{i}}$. p $\left(T_{e_{m,}}\right)$ is the capability of transmission by the transmitter’s m $T_{e_{m,}}$ on edge $e_{m}$, and $\lambda_{i}$ Is the background noise present at the “receiver node” $r_{e_{i}}$. Then, the “SINR” for the edge is defined as follows:(3)$$ẟ(e_{d_{i}})\;=\;\frac{G\left(T_{e_{m,}}{\;r}_{e_{i}}\right)p\left(T_{e_{m}}\;\right)\;}{I_{f}\;\left(e_{d_{i}}\right)}$$

According to condition (3) above, when $I_{f}$($e_{d_{i}}$) increases, the transmission power *p*($T_{e_{m,}}$) needs to increase proportionally to maintain the same “SINR” value on the connection. However, when *p*($T_{e_{m,}}$) expands, various topology connections might experience more interference. To maintain constant signal strength and communication quality, these connections must strengthen their “transmission capacity [22]”. This could shorten the system’s lifetime and increase the nodes’ energy consumption.

### 3.3. Factor for Congestion Level

The “cross-layer data” are transmitted as a “state frame” following the construction of the numerous nodes connecting the source and sink. This “frame” is sent “upstream” to update and communicate the “node’s congestion” data with other nodes. Congestion levels are indicated for each node by
(4)cr=tsse
where $s_{e}$ stands for service rate and $t_{s}$ Stands for the rate of incoming traffic. The quantity of the packets that flow into “the physical layer of the protocol stack” within a given period is known as a node’s rate of incoming traffic. The number of packets streamed down to the channel in a given time is also referred to as the service rate [8,11,23].

**A.** 
**Choosing Factor for Cluster Heads**


Each node's “cluster head choosing factor (CCF)” determines which routing table node will be the next-hop node. CCF is a function that takes into account three variables: the congestion level $C_{l}$ at the next hop, the SINR value of the link έ($e_{i}$) between the current and next-hop nodes, and the “survivability factor”. $P_{s}$ of the path to “the destination through that next hop”. The CCF is given as
(5)$$CCF=\left(\alpha *έ\left(e_{i}\right)\right)+\left(\beta *P_{s}\right)+\left(\gamma *\left(1-C_{l}\right)\right)$$

The three components έ($e_{i}$), $P_{s}$, and $C_{l}$ The CCF are given different weights, in this case, using the values  α, β, and γ. In the cluster head selection, the intensity of these three components can be selected based on the requirement. The simulation is carried out in terms of three “weighting coefficients” as α =β=γ=13, to demonstrate an equivalent influence from each PCF component. The following goals are reached by standardizing the values as follows:(6)α+β+γ=1

According to condition (6), the input for the “adaptive fuzzy c-means clustering” will be the “CCF factor”. Finally, this task is completed using the three factors, including “SINR”, “congestion level”, and “survivability factor” in the network of clustered “sensor nodes” [14,18].

**B.** 
**Algorithm for adaptive fuzzy c-means clustering based on CCF.**


The system nodes cluster head is selected among the nodes with the highest significant “SINR”, “congestion level”, and “survivability”. Utilizing the “adaptive fuzzy c-means (AFCM)” “clustering algorithm”, the sensor nodes are organized into groups. Support kernel matrices are constrained in this situation by purposefully using the CCF factor in clustering [24,25,26]. There are numerous initial cluster centers in this algorithm. The AFCM algorithm uses fuzzy memberships to distribute the information data for each class.
(7)$${\tilde{J}}_{\sigma n}=\sum_{l=1}^{L}\sum_{m=1}^{M}{\left(v_{ij}\right)}^{n}\frac{\left\|{\hat{S}}_{l}-q_{m}{}^{2}\right\|}{CCF_{l}}$$

${\tilde{S}}_{l}$ indicates the support value, $q_{m}$ indicates the $m^{th}$ Cluster center and n indicate the constant esteem in condition (4), Where CCF shows that the cluster head is selecting a cluster l factor, mentioned in condition (5), “the membership function” expresses the possibility of a pixel’s belongingness into a target cluster [27,28,29]. The conditions (8) and cluster centers update the “membership functions” and “cluster centers” (9).
(8)v¯lm=1∑k=1q‖S˜l−qmσ¯l‖‖S˜l−qkσ¯l‖2n−1

The condition is used to process the cluster’s centroid (9),
(9)$$z_{m}^{'}=\frac{\sum_{l=1}^{L}{\overset{¯}{v}}_{lm}^{n}.{\tilde{S}}_{l}}{\sum_{l=1}^{L}{\overset{¯}{v}}_{lm}^{n}}$$

This calculation is repeated until the coefficients’ difference between two cycles is approximately at the predetermined limit.
(10)$$\max_{\mathrm{lm}}{\|\overset{¯}{V}}_{lm}^{\left(k\right)}-{\overset{¯}{V}}_{lm}^{\left(k+1\right)}\|<\psi$$

Condition (8) ψ is a number between 0 and 1. Computation is carried on until efficient clustering is achieved. In Algorithm 1, this AFCM clustering is indicated [30].


**Algorithm 1:** A “CCF-based adaptive fuzzy c-means clustering” algorithm**Input:** input $N_{O}=\{N_{O_{1}},\;N_{2},\;N_{O_{3}}\;\ldots .\;N_{O_{i}\}}$ the group of network nodes, SINR, level of congestion, and survivability
**Output:** clustered data**Begin**
   **for**
 j= to N0**.** do       For belonging to cluster I, Node j. receives coefficient vij    **End for**       Repeat    **for** i=1 to k. do          Using condition (8), determine each cluster’s centroid (9)    **End for**    Repeat from Begin: *Until the stop, the condition is met***End**



After the nodes have been clustered, it is assumed that the clustered nodes forward the packets. The “AQLG operation” is performed on each packet before the retransmission or rebroadcasting [9].

**C.** 
**Using DNA-BASED Encryption to Secure Data Packets**


Our proposed scheme, the “DNA-based encryption technique” for secure routing for Zigbee devices in IoT networks, is a novel “DNA block cipher symmetric algorithm”. The “DNA sequence” generates a secret encryption key (SecKey). This proposed technique perfectly fits the limited “computation resources” of “Zigbee devices [10,31,32]”. The proposed “encryption technique” performs the following two steps: (1) key generation and (2) encryption.

Generation of the Key:

The “DNA sequence” generates the keys regardless of size, where each letter in the DNA sequence is followed by four randomly arranged bits. DNA sequences are made up of a series of letters. In Table 2, each alphabet is represented in binary using two bits. The four “DNA letters (A, C, G, and T)” are then used to segment the DNA sequence into 1-byte segments ($\{K_{O_{1}}$, $K_{2}$, $K_{O_{3}}$ … $K_{O_{i}}$), where these segments stand in for the secret key (SecKey).

Process of encryption:

The suggested “DNA-based” technique is a “block cipher” that performs operations on the input data bytes, including substitution and transposition. Figure 2 depicts the detailed flow of DNA-based encryption techniques. The two procedures are explained below:

1. Phase of Substitutions: The input data must be divided into N segments ($\{B_{O_{1}}$, $B_{O_{2}}$, $B_{O_{3}}$ … $B_{O_{i}}$) with a 1-byte segment size before applying the substation phase. Using the XOR operation between DNA segments of Sk ($\{K_{O_{1}}K_{O_{2}}$, $K_{O_{3}}$ … $K_{O_{i}}$),) and I bytes ($\{B_{O_{1}}$, $B_{O_{2}}$, $B_{O_{3}}$ … $B_{O_{i}}$),) of the input data segments, the updated bytes (A1, A2,…, An) are generated, and the process is repeated “N” times. Finally, the “Ai” bytes are generated as a result.

2. Phase of transposition: The resultant Ai bytes’ bits are switched during the transposition phase. Table 3 represents the DNA sequences and the set of operations that are used to generate the Sk bytes, which are the foundation for the “bits-swapping process” carried out on A $\{A_{O_{1}}$, $A_{O_{2}}$, $A_{O_{3}}$ … $A_{O_{i}}$),). Do not swap, for instance, if the first two-bit sequence of the alphabet is “00” and “01”. The DNA sequence’s third and fourth bits are swapped if they are “01” and “11”. Figure 3 depicts the transposition flow. Likewise, until the entire data gets encrypted, the “substitution” and the “transposition” operations are performed repeatedly [33].

Encryption Process:

Transposition is performed first, then substitution, when the “decryption” initializes. The “final byte of the encrypted input data”, or the Edata, serves as the starting point for the decryption process, which proceeds gradually to the first byte [12,34,35]. Given that both the sender and the receiver of the input data must follow the “swapping rules” in Table 3, the “transposition phase restores” the “swapped bits” to their original positions. After the “transposition phase”, “the substitution phase” starts by performing an XOR operation using the final byte of the “DNA sequence”, in which the transmitter of the Edata terminates the “substitution operation” and the last byte of the Edata. Up until the first byte of Edata, the substitution is still in effect. The initial input data will be restored after the substitution phase.

**D.** 
**Utilizing adaptive krill herd optimization for secure routing**


The “adaptive krill herd (AKH)” algorithm acquires effective routing to transmit the data packets. Figure 4 depicts the detailed flow. This “optimization algorithm” chooses an uncongested path for data flow [13,36]. The natural krill herd phenomenon requires this iterative heuristic approach. This is mainly employed to address optimization issues. The “krill herd optimization” pseudocode is found in Algorithm 2. **Algorithm 2:** Pseudocode for “the krill herd optimization algorithm”.BeginDefine the population size (S') and the loop iteration (I^max ):Initialization Phase:Set the initial sequence I'=1;Set up the population data and cluster information as input.S˜=1,2,3,…..S' Of krill arbitrarily.**Fitness assessment:**Analyze each krill following the location’s instructions.While I'<I^max doSort the population or krill from best to worst.**For**
i=1:S' doCalculate the three motions,
**(1)**
*The krill drive action***(2)**
*Hunting behavior***(3)**
*Physical agglomeration*The krill location in the inquiry space should be updated.Consider each krill in light of its proximity.End For i
Sort the current krill from best to worst, then identify the “best” ones with optimal metrics.I^max=I'+1. End WhilePredict (“The krill’s finest result”).End

The described “krill herd optimization” successfully chose a not congested path using the earlier steps [14].

Step 1: Initializing standardized data sets the optimization process in motion.

Step 2: Fitness is evaluated based on the adaptable krill individuals’ positions. This adaptive method can preserve a safe distance from surrounding minima, accelerate convergence, and decrease the computation necessary to reach the optimal solution. The K.H. “adaptive methodology” is described as follows:(11)Xii+1 =Xii +Rn ×1tbestfitt−fittbestfitt−worstfitt where $R_{n}$ is an arbitrary number,$X_{i}^{t+1}$ is a new dimension solution in the $t^{th}$ iteration, and f(t) is the fitness value.

Step 3: The krill are first arranged from best to worst in the original version.

Step 4: After that, each krill’s movement updates are controlled according to the present circumstances.

The search updates are completed using (12)$${\overset{¯}{F}}_{z}\left(\hat{t}+1\right)=S_{f}\beta_{x}+\omega_{i}{\overset{¯}{F}}_{z}\left(k'\right)$$
(13)$$\beta_{z}=\beta_{z}^{food}+\beta_{z}^{best}$$ where $S_{f}$ denotes the foraging speed, $\omega_{i}$ the inertia weight, and the optimal outcome of the “$z^{th}$ krill” individual.The updated “induced movement” associated with information thickness preservation is depicted as follows: (14)$${\overset{¯}{M}}_{z}\left(\hat{t}+1\right)=\overset{¯}{M}{\hat{{\overset{¯}{z_{l}}}_{z}}}_{max}$$
(15)$$\alpha_{z}=\alpha_{z}^{total}+\alpha_{z}^{t\arg et}$$

The best outcome for the $z^{th}$ krill represents the most extreme activation speed and the close influence $\alpha_{z}^{t\arg et}$ of the individual krill on its neighbors.

c.“Final movement update” depicts and synchronizes the distribution of physical nodes, (16)$${\overset{¯}{D}}_{y}\left(\hat{t}+1\right)=\overset{¯}{D}{\frac{1-i}{i_{max}}}_{max}$$ where ${\overset{¯}{D}}_{max}$ Denotes the most excellent “diffusion speed”, and δ represents the “random directional vector between”−1 and 1.

Step 5: According to recently demonstrated advancements, a node’s location $y^{th}$ is determined by considering the specific development of the time-related parameters and the location of the krill during $\hat{t}+\Delta \hat{t}$.
(17)$${\overset{¯}{K}}_{z}\left(\hat{t}+\Delta \hat{t}\right)={\overset{¯}{K}}_{z}\left(\hat{t}\right)+\Delta \hat{t}\frac{d{\overset{¯}{K}}_{z}}{d\hat{t}}$$
where $\Delta \hat{t}$ It stands for a fundamental constant. The “krill individual’s” location is updated using the reference condition, and the optimal result is attained.

Step 6: The pausing equation makes sure the functional evaluation is complete. The best node individual site is identified until the stopping stage has been achieved, and “the krill population” is rated from higher to lower performance. The flow chart for managing “krill herds” is shown in Figure 4.

## 4. Results and Analysis

To evaluate the performance of the suggested simple “DNA-based encryption” method for Zigbee devices. We put the proposed algorithm into practice. We evaluated its performance in comparison to state-of-the-art encryption algorithms such as “DES and AES”, which are considered industry standards in terms of “key size”, “encryption time”, and distortion percentage. The DES, AES, and proposed DNA algorithms’ key sizes are shown in Table 4 and Table 5.

The implementation of our proposed “efficient routing” technique in a Z.N. for “IoT applications” was performed on “MATLAB 2018a”. We evaluated the proposed scheme in terms of multiple factors, such as “packet delivery ratio”, “energy consumption”, “packet drop”, and “remaining energy”. The comparative evaluations were performed with “directed diffusion routing pro-energy-aware routing protocol (SGEAR)” and “survivable path routing (SPR) protocols”.

Our proposed work is an efficient and effective routing method in a Z.N. for IoT applications that requires the working stage of MATLAB 2018a. Different execution estimates are compared to “the current directed diffusion routing protocol”, “sub-game energy-aware routing protocol (SGEAR)”, and “SPR protocols” for estimating the performance of the proposed scheme. The following list contains the simulation parameters used in the suggested routing protocol. The Simulation Parameters values show in Table 6.

Remaining Energy: Ten information packets are started by source nodes in the system every second and sent to the destination node over multiple hops. Figure 5, Figure 6 and Figure 7 depict the energy levels between the source and destination Zigbee devices during the various rounds of packet transmission.

Comparing the proposed routing with the current “Survivable Path Routing (SPR)”, the “Zigbee network nodes” using the “directed diffusion protocol” result in higher energy strength novelty, as shown in Figure 5, Figure 6 and Figure 7. The “energy capabilities” of the “Zigbee network nodes” are similar to the proposed protocol. Therefore, extending the network connection might be advantageous and moderately increasing the system’s resilience. The proposed “protocol’s maintenance phase” enables the relay nodes to transfer each data packet after confirming it is under a current energy threshold. The pathways are reconfigured if any node’s residual energy limit drops within the cutoff. In the same way, the “path selection metrics” are calculated and processed. Finally, network connectivity will be improved because each node’s battery capacity will remain constant.

## 5. Conclusions

This paper proposes a lightweight encryption method that utilizes the randomness of DNA sequences. Key operations, called “substitution” and “transposition”, are performed on the DNA sequence to achieve encryption. The reasonable key size and the randomness ensure the proposed algorithm’s robustness, and the key data transfer routing is made more secure using DNA encryption. The key generation and encryption processes met the limited resources regarding processing and memory size for Zigbee devices. The simulation results demonstrate that the proposed DNA-based encryption technique performs better than the alternatives in high-traffic networks with low energy consumption.

## Figures and Tables

**Figure 1 life-13-01147-f001:**
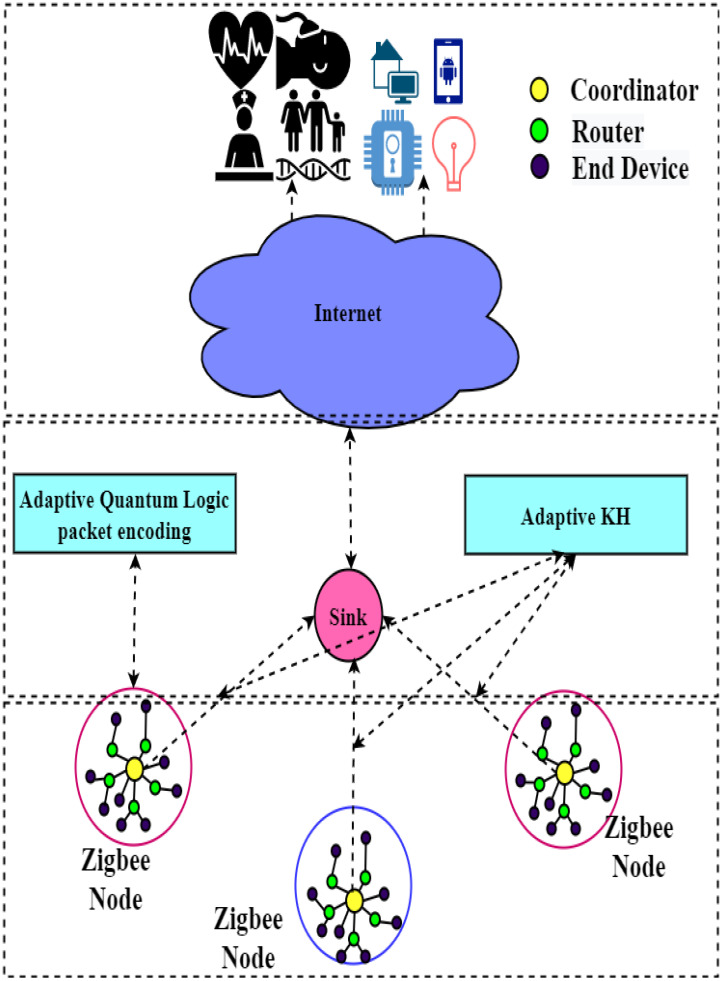
The proposed framework for DNA-based Encryption in Zigbee Network.

**Figure 2 life-13-01147-f002:**
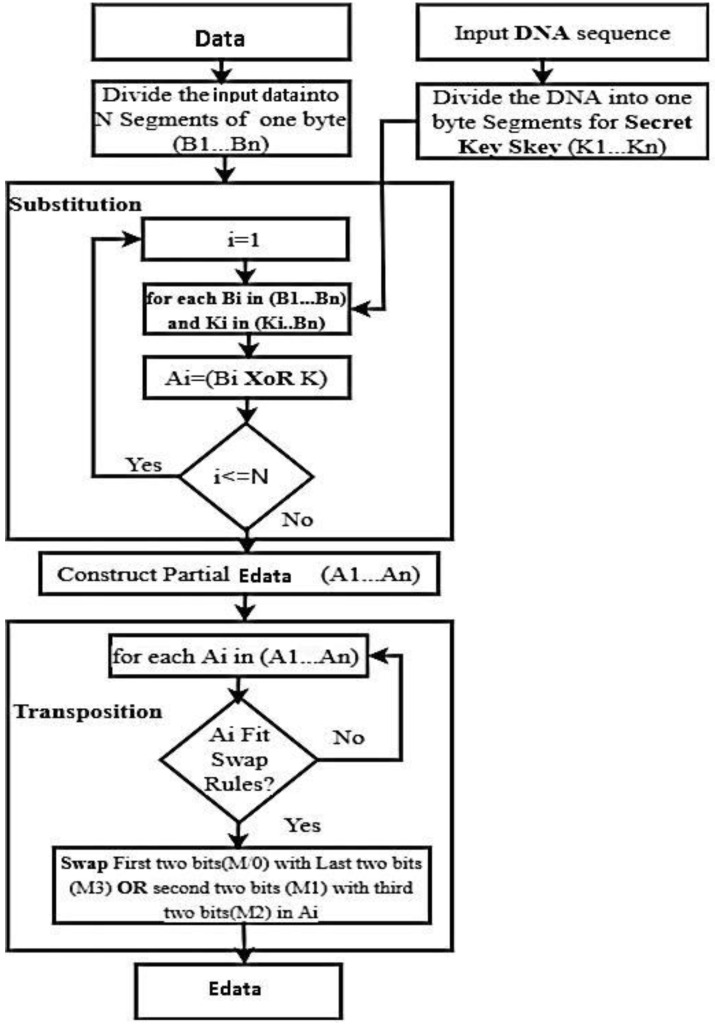
The proposed “DNA-based Encryption Process”..

**Figure 3 life-13-01147-f003:**
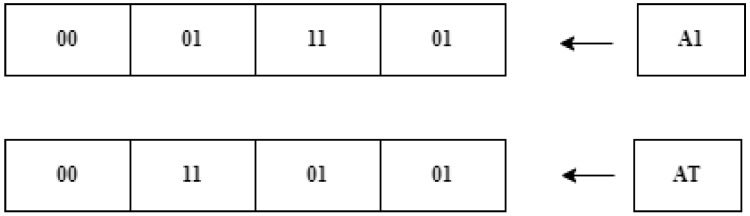
Transposition example.

**Figure 4 life-13-01147-f004:**
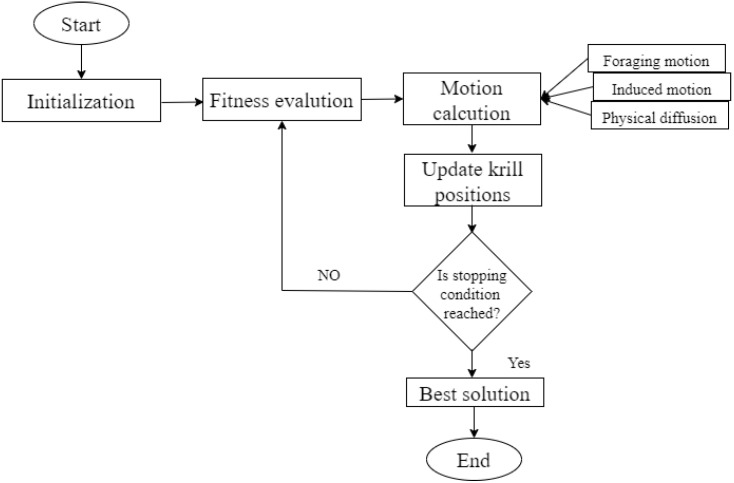
Adaptive “KRILL HERD OPTIMIZATION” flow diagram.

**Figure 5 life-13-01147-f005:**
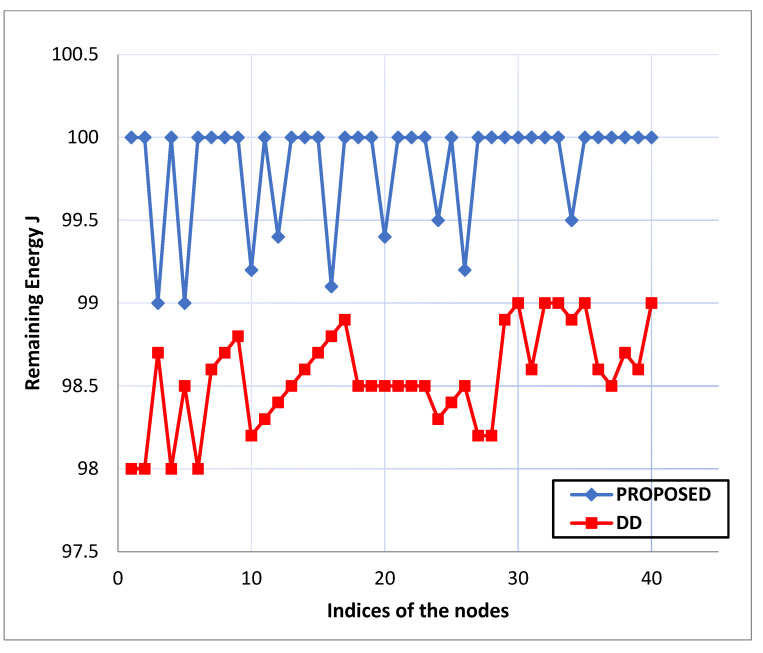
Comparison graph of remaining energy for proposed routing with existing “Directed Diffusion (D.D.)”. Routing Protocol.

**Figure 6 life-13-01147-f006:**
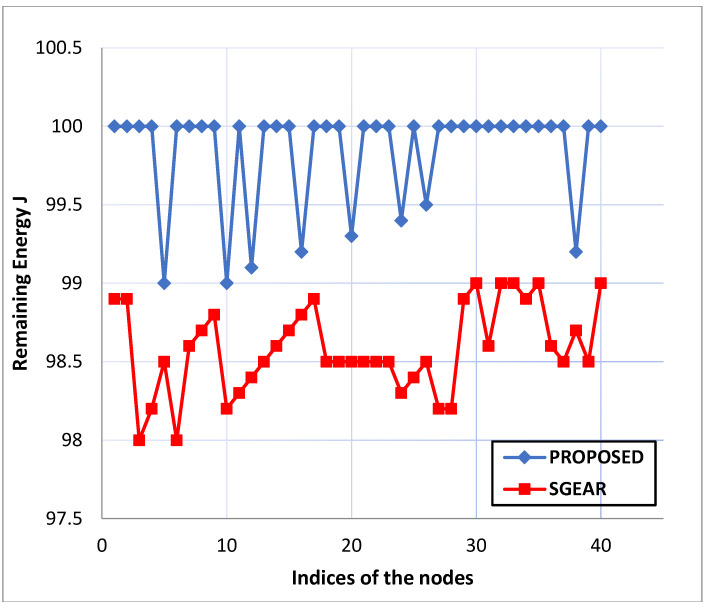
Comparison graph of remaining energy for proposed routing with existing “Sub-Game Energy-Aware Routing Protocol (SGEAR)”.

**Figure 7 life-13-01147-f007:**
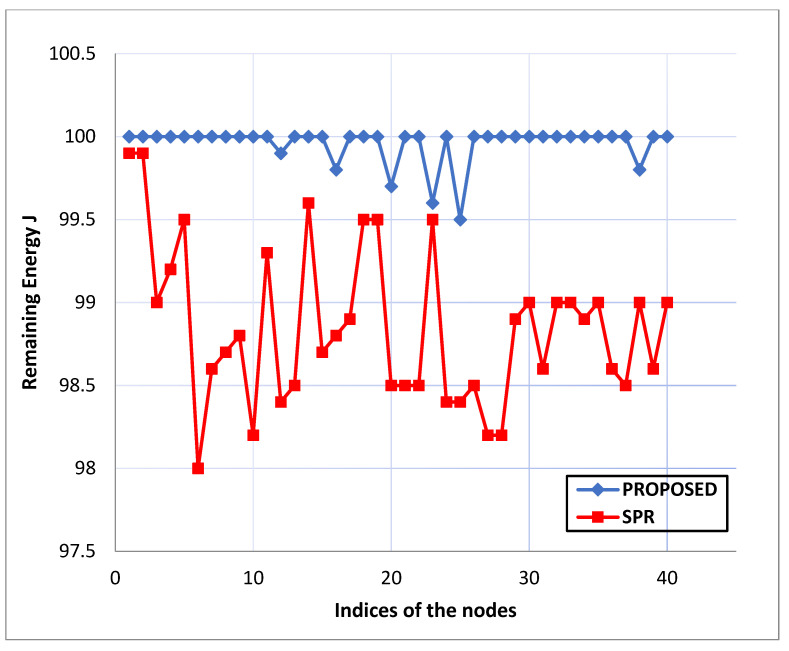
Comparison graph in terms of remaining energy.

**Table 2 life-13-01147-t002:** The letters in the DNA sequence are represented in binary.

S. No.	Letters from DNA	Representation in Binary
1	A	00
2	T	01
3	C	10
4	G	11

**Table 3 life-13-01147-t003:** Swapping rules.

S. No.	The Letter on DNA Sequence 1	The Letter on DNA Sequence 2	The Intended Operation
1	A	T	“Do not swap”.
2	T	T	“Do not swap”.
3	C	G	“Swap”.
4	T	G	“Swap”.
5	G	G	“Swap”.
6	A	T	“Do not swap”.

**Table 4 life-13-01147-t004:** DES, AES, and proposed DNA algorithms’ key sizes.

Algorithm for Encryption	Scope of the Key
“DNA algorithm”	“8” bits
“AES”	“256” bits
“DES”	“56” bits

**Table 5 life-13-01147-t005:** Time consumed for encryption.

The Current Image	Time Taken by the Proposed DNA-Based Scheme	Time Taken by “DES”	Time Taken by “AES”
“POOL”	“203.125” s	“2625” s	“2609.375” s
“PETRA”	“62.5” s	“1421.85” s	“1421.875” s
“AQSA”	“109.375” s	“2093.75” s	“2125” s

**Table 6 life-13-01147-t006:** Simulation Parameters.

Parameter Name	Parameter Value
Propagation mode	Shadowing model
Transmitting range	40 m
MAC Protocol	IEEE802.15.4
Traffic Flow	Constant Bit Rate
Data transfer Rate	Ten pkt/sec
Packet size	50 bytes
Initial energy	50 bytes
Cycle time	10 s

## Data Availability

Not applicable.

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
