# Peer review of "Efficient Secure Communication in Zigbee Network Using the DNA Sequence Encryption Technique"

_life, 2023, doi:10.3390/life13051147_

Round 1

Reviewer 1 Report

In the manuscript entitled Efficient Secure Communication in Zigbee network using DNA Sequence Encryption Technique, the authors claim that they have proposed a method that uses DNA sequence randomness, substitution, and transposition to generate a complete secret key for Zigbee computing resources. The work is novel as the lightweight encryption algorithm is suitable for Zigbee IoT devices that have limited computing resources, including processing power and storage capacity. However, the manuscript has several sweeping statements without references and some part of the write-up is non-scientific.  

For example,

In abstract: Pattern matching is required as part of the pattern discovery process in today’s "culture to"  detect structural and functional behavior in genes.

page 1, line no 30: DNA sequencing requires an effective and quick algorithm because DNA sequences create large databases.

page 2, line 50: It contains directions for generating every protein in our bodies, much like a "recipe book".

page 2, line 73: Diseases are caused by duplicating genes, repetitive genes, deleted genes, and the precise presence of disease-affected genes, among other factors.

In particular, the statements written on DNA are mostly wrong or incorrect.  Of note, the idea of DNA sequence substitution/ base pairing has been implemented correctly. I suggest that these computational researchers do a rigorous literature survey and get molecular biologists' help to improve their manuscript.

Author Response

Reviewer 1

Point 1: Pattern matching is required as part of the pattern discovery process in today’s culture to detect structural and functional behavior in genes.

Response 1: We are not concentrating on pattern matching in this paper. Our main focus is on the security provided in the ZigBee network. Pattern matching is required to find the pattern and structures in DNA sequence, but we are not using pattern matching algorithms. Our main goal is to provide security with effective and efficient algorithms.

Point 2: DNA sequencing requires an effective and quick algorithm because DNA sequences create large databases.

Response 2: Regarding DNA sequence in this paper, we are not considering database-related tasks. DNA sequences create large databases, and finding patterns and structures in those databases takes more time, requiring efficient DNA sequence algorithms. Because of our primary goal of security, we are not concentrating on database-related tasks.

Point 3: It contains directions for generating every protein in our bodies, much like a recipe book.

Response 3: DNA sequence contains directions and other attributes. Protein is a biological term. From our security perspective, we will not be considering these terms. Using the DNA sequence approach, we can create and apply effective and efficient security keys for providing strong security in the Zigbee network.

Point 4: Diseases are caused by duplicating genes, repetitive genes, deleted genes, and the precise presence of disease-affected genes, among other factors.

Response 4: These are biological terms we are not using in this paper. Our primary motto is to provide efficient security in low-powered Zigbee devices. The DNA sequence approach is simple and can generate a strong security key. As a result, we offer a strong security algorithm. It is impossible to break the security key.

Reviewer 2 Report

Dear authors, first of all congratulations for your interesting research.

I have few suggestions:

1. Please include some ethics statements, for you are working with DNA sequencing, and therefore I believe it should mandatory in your case.

2. Please, perform an extensive language check. There are several sentences which are cut in the middle and are not clearly written. Just as an example (there are many others across the text): The pausing equation makes sure the functional evaluation is complete. No matter whether the stopping stage has not yet been achieved, identify the best node individual site and rate “the krill population” from the higher performance to the lower.

Author Response

Review 2:

Point 1: Please include some ethics statements, for you are working with DNA sequencing; therefore, I believe it should be mandatory in your case.

Response 1: We utilized the DNA sequence's random nature to generate a robust and efficient secret key. The generated secret key is impossible to decode due to its random nature. It's not possible to crack the secret key.

DNA encryption is preferred instead digital encryption because most cryptographic techniques have been cracked partially by the new computer generation, such as Quantum Computing [18]. Moreover, regarding the key generation issue, the randomness and complexity of DNA sequence attached an additional layer of security for DNA-based encryption methods [19]. The DNA sequence consists of four alphabets (A, C, G, and T), and each alphabet is associated with a nucleotide. The DNA sequence is usually quite long, and the publicly available DNA sequences are to be around 55 million. The DNA sequence is mainly used for a secret key generation where it must only be known by the sender and the receiver [20].

We are proposing a new lightweight encryption algorithm based on the DNA sequence that fits the computation resources of IoT devices. The key generation of the proposed algorithm is completely random based on the DNA sequence, which makes it very difficult to break. Moreover, the generated key is used to make simple,

logical and strong confusion and diffusion on the plain text based on the random nature of the DNA sequence and which satisfies IoT computation capabilities.

  1. Fernández-Caramès, T.M., Fraga-Lamas, P.: Towards post-quantum blockchain: a review on blockchain cryptography resistant to quantum computing attacks. IEEE Access 8, 21091–21116 (2020)

  1. Omran, S.S., Al-Khalid, A.S., Al-Saady, D.M.: A cryptanalytic attack on Vigenère cipher using genetic algorithm. In: 2011 IEEE Conference on Open Systems, pp. 59–64. IEEE(2011)

  1. Barman, P., Saha, B.: DNA encoded elliptic curve cryptography system for IoT security. Int. J. Comput. Intell. IoT 2, 478–484 (2019)

Result:

The first evaluation metric is the key size used in the proposed DNA-based encryption algorithm. To achieve high protection for the encrypted text, the key must be large and entirely random to be unbreakable by attackers. To overcome the computation resources of IoT devices, the key size for the proposed DNA-based algorithm for the encryption process is 8 bits, as shown in the Table below.

The key size of the proposed DNA algorithm, DES, and AES

Encryption System

Size of the key

DNA Algorithm

8

AES

256

DES

56

The second metric, encryption time, usually plays a significant role in different communication applications and its related encryption algorithms, especially for IoT devices and applications. In the proposed DNA algorithm, logical XOR substitution and transposition rules operations reduced the time needed to encrypt plain text (time in milliseconds ms). This means less processing time and memory that fit IoT computation resources compared to other algorithms. As shown in the table below.

Encryption Time

Text

Proposed DNA Algorithm

DES

AES

text1

203.125

2625

2609.375

text2

62.5

1421.85

1421.875

text3

109.375

2093.75

2125

Conclusion:

This paper proposes a lightweight encryption method that utilizes the randomness of DNA sequences. The key operations, called "substitution" and "transposition," are performed on the DNA sequence to achieve encryption. The reasonable key size and the randomness ensure the proposed algorithm's robustness and strong, and the key data transfer routing is made more secure using DNA encryption. The key generation and encryption processes meet the limited resources in terms of processing and memory size for Zigbee devices. The simulation results demonstrate that the proposed DNA-based encryption technique in high-traffic networks performs better than the alternatives. It has a short end-to-end delay, a high packet reception rate, and low energy consumption.

Round 2

Reviewer 1 Report

I think the manuscript is well-written and has merit to be published. However, I suggest these authors establish the need for this review as many recent excellent reviews are available in the past 2 years.
